# Nanoemulsified Formulation of *Cedrela odorata* Essential Oil and Its Larvicidal Effect against *Spodoptera frugiperda* (J.E. Smith)

**DOI:** 10.3390/molecules27092975

**Published:** 2022-05-06

**Authors:** Ana Sofía Lemus de la Cruz, Josefina Barrera-Cortés, Laura Patricia Lina-García, Ana C. Ramos-Valdivia, Rosa Santillán

**Affiliations:** 1Centro de Investigación y de Estudios Avanzados, Departamento de Biotecnología y Bioingeniería, Instituto Politécnico Nacional (Cinvestav-IPN), Unidad Zacatenco, Av. Instituto Politécnico Nacional 2508, Col. San Pedro Zacatenco, Ciudad de México 07360, Mexico; asumel16@gmail.com (A.S.L.d.l.C.); aramos@cinvestav.mx (A.C.R.-V.); 2Centro de Investigación en Biotecnología, Universidad Autónoma del Estado de Morelos, Av. Universidad 1001, Chamilpa, Cuernavaca 62209, Mexico; lina_g63@yahoo.com; 3Departamento de Química, Cinvestav-IPN, Unidad Zacatenco, Av. Instituto Politécnico Nacional 2508, Col. San Pedro Zacatenco, Ciudad de México 07360, Mexico; rsantill@cinvestav.mx

**Keywords:** botanicals, fall armyworm, sustainable insecticides, wood waste

## Abstract

*Cedrela odorata* L. is a plant species from the Meliaceae family that is cultivated for timber production. Although the *C. odorata* essential oil (EO) contains mainly sesquiterpenes, its insecticidal potential is unknown. The lipophilic properties and high degradation capacity of EOs have limited their application for use in pest control. However, the currently available knowledge on the nanoemulsification of EOs, in addition to the possibility of improving their dispersion, would allow them to prolong their permanence in the field. The objective of the present work was to develop a nanoemulsion of the *C. odorata* EO and to evaluate its larvicidal activity against *Spodoptera frugiperda*. The EO was obtained by the hydrodistillation of *C. odorata* dehydrated leaves, and the nanoemulsion was prepared with non-ionic surfactants (Tween 80 and Span 80) using a combined method of agitation and dispersion with ultrasound. The stability of the nanoemulsion with a droplet diameter of <200 nm was verified in samples stored at 5 °C and 25 °C for 90 days. Both the *C. odorata* EO and its corresponding nanoemulsion presented lethal properties against *S. frugiperda*. The results obtained provide guidelines for the use of wood waste to produce sustainable and effective insecticides in the fight against *S. frugiperda*. In addition, considering that a phytochemical complex mixture allows the simultaneous activation of different action mechanisms, the development of resistance in insects is slower.

## 1. Introduction

*Spodoptera frugiperda* (J.E. Smith) (Lepidoptera:Noctuidae) is an economically important pest due to its great capacity for migration and adaptation, as well as its ability to develop resistance to pesticides [1,2,3,4]. In its larval state, *S. frugiperda* affects a wide variety of crops, of which corn, sorghum, rice, cotton, sugar cane, peanuts, Bermuda grass, and soybean stand out [5,6,7]. To control this type of pest, it has been common practice to apply broad-spectrum synthetic chemical insecticides, such as pyrethroids, organophosphates, and carbamates [8,9,10]. However, due to the problems of environmental contamination and toxicity to humans and non-target organisms, the development of sustainable and effective insecticides against *S. frugiperda* is an initiative that has been enforced over the last few decades [3]. Cry protoxins produced by different species of bacteria of the genus *Bacillus* have been applied for use in the control of *S. frugiperda* [3]. However, the inclusion of protoxins in corn and cotton plants has accelerated the development of resistance to this pest [11]. Research to combat *S. frugiperda* is currently focused on botanical insecticides: extracts and essential oils [12]. It is assumed that the wide variety of phytochemicals they contain, along with their different modes of action, retard the development of resistance in insect pests.

Essential oils (EOs) are secondary metabolites of plants that, due to their aromatic properties and molecular composition, have a wide variety of applications mainly in the pharmaceutical, food, and agricultural sectors [13,14,15]. In the agricultural sector, its properties as repellents, insecticides, and growth reducers have been used to control a wide variety of insects [16]. The mechanism of action of EOs involves neurotoxic effects in various modes of action, such as the inhibition of acetylcholinesterase (AChE), the functionality alteration of gamma-aminobutyric acid (GABA) receptors, and as an octopaminergic agonist [14]. With some exceptions (azadirachtin), unlike botanical extracts, whose regulation remains restricted due to the variable composition of plant extracts and the high toxicity of some phytochemicals (rotenone, aconitine, and nicotine) [17,18,19,20], pest control based on EOs for human consumption is not subject to regulation because it involves easily degradable products that have low toxicity for non-target organisms [21]. The drawback of EOs is their high volatility, easy degradation, and insolubility in water. However, the success of encapsulating the oil in nanoemulsions has made it possible to extend its viability in the environment, and even control its dosage [14,22].

In a nanoemulsion, EO is dispersed in an aqueous fluid, where the micelle size is controlled by a surface-active agent that surrounds each oil droplet. The homogeneous dispersion allows a uniform application of the oil on the plant surface and, thanks to the lipophilic properties of the oil, this adheres to the plant or infiltrates the insect cuticle [23]. He et al. (2016) [24] studied the biological activity of an insecticide in terms of the droplet size of the dispersed fluid and the drop number per unit area of the plant leaf. These authors found that the smaller the EO droplet size, the more uniform its application, and the more effective its biological activity. In addition to maintaining the droplet size of the oil in the nanoemulsion, the surfactant agent prevents EO evaporation and hydrolysis and oxidation phenomena, which maintains the chemical stability of the EO. This chemical stability allows for the essential oil to be kept viable for long periods [25]. The application of insecticides in nano-type formulations (nanoemulsions or nano-suspensions) not only guarantees the effectiveness of the insecticides (whether synthetic, biological, or botanical), but also reduces environmental pollution problems thanks to their better use and dispersion [26].

Appendix A presents a summary of the biological activity of the EOs studied for combating *S. frugiperda*. The reported lethality is highly variable, which is evident not only in the particularities of each oil, but also in how they were applied [24]. It has been reported that EOs are more effective in non-emulsified formulations, and there is currently extensive research on the development of this type of formulation [27]. A nanoemulsion can be prepared by high or low-energy methods. High-energy methods are widely used industrially and involve agitation operations with high shear forces. This method requires special equipment, such as ultrasound, homogenizers, and microfluidiz-ers. Low energy methods depend on the internal energy of the components. The latter methods are based on the change of phases from the effect of an increase in temperature or an increase in the concentration of one of the components [22].

*Cedrela odorata* L. is a leafy plant grown for lumber manufacturing. The peculiar aroma that emanates from its leaves and flowers has given rise to numerous reports on its chemical composition for exploitation mainly in cosmetics [28,29,30,31,32,33,34]. The *C. odorata* EO is mainly made up of monoterpenes, sesquiterpenes, and phenylpropanoids, and even though compounds from these groups have insecticidal properties, there is little information on their application in pest insect control [35,36]. The objective of this work is the development of a nanoemulsified formulation of the *C. odorata* EO. *Cedrela odorata* belongs to the Meliaceae family, to which the neem tree also belongs, a plant known for its insecticidal properties that is already approved for use in insect pest control applications [37,38]. Considering that large volumes of waste arise from the wood production process, we envisaged that the availability of botanical insecticides derived from *C. odorata*, in addition to helping solve the problem of crop loss, could contribute to the development of sustainable insecticides.

## 2. Results and discussion

### 2.1. Essential Oil Extracted from Leaves of Cedrela odorata

The physical properties of the EO extracted from fresh and dehydrated leaves of *C. odorata* are presented in Table 1. The data do not show differences in the physical properties because of the drying treatment. However, the extraction yield increased by 126% when using leaves that were previously dehydrated. The higher extraction yield that results when using dehydrated leaves has already been reported, and these results could explain why most of the studies carried out with phytochemicals use dehydrated leaves [39]. The lower extraction yield that results when fresh plants are used could be associated with the higher binding strength of the oil molecules in the moist plant tissue [40]. In addition to the moisture content in the vegetables, the extraction method is another factor that determines the yields and composition of the extracted oils. At present, there is knowledge of innovative methods that have made it possible to increase yields [13]. However, they can present disadvantages, such as high extraction cost (supercritical fluid extraction), the potential formation of free radicals (ultrasound-assisted extraction), and the formation of undesirable compounds (microwave-assisted extraction). Although hydrodistillation can lead to the evaporation loss of polar compounds, as well as to the alteration of the oil chemical structure, it is a relatively cheap method and easy to implement.

Although plants with the ability to produce essential oils have been identified, extraction yields are usually low, less than 1% [31,41]. Maia et al. (2000) [42], who applied the hydrodistillation method to extract EO from *C. odorata* dehydrated leaves collected from Sao Carlos, Brazil, reported an extraction percentage of 0.23%. A similar extraction yield of 0.19% was reported by Asekun and Ekundayo (1999) [31]. These authors used *C. odorata* leaves collected from the plantations of the Forest Research Institute of Nigeria (FRIN), Ibadan. The leaves were dehydrated with air and the essential oil was extracted using hydrodistillation for 4 h. Lower yields of 0.1% were reported by Martins et al. (2003) [43], although in this case, the authors extracted EO from the bark of *C. odorata* trees growing in the Central African nation of São Tomé and Príncipe. Compared with the extraction yields reported by these authors, the yields obtained in the present work are higher.

### 2.2. Composition of the C. odorata Essential Oil

A chromatographic analysis of the EO extracted from the fresh and dehydrated leaves of two batches of *C. odorata* leaves allowed us to identify between 12 and 15 compounds, mostly sesquiterpenes, that were present in a concentration higher than 1% (Table 2 and Appendix A). The mass spectrum of the identified compounds was compared to the mass spectrum of the NIST library. In both mass spectra, the predominant ions coincided and the match level allowed the identification of the major compounds with an assertiveness > 90% (Figure 1): germacrene D was identified with an approximation percentage of 97.3% and the caryophyllene with a percentage of 96.9%. Mass spectra similar to those obtained in the present work for the main compounds have been reported by Noge and Becerra (2009) [44] and Duarte et al. 2010 [45]. Germacrene D and caryophyllene were the compounds extracted in the highest concentration, in percentages of 10–37% and 24–52%, respectively. The presence of sesquiterpene hydrocarbons and sesquiterpene alcohols in the EO of *C. odorata* leaves has already been reported by other researchers, as has the predominance of between two and five compounds. Although germacrene D and caryophyllene are on the list of major compounds found in *C. odorata*, they have not been extracted in the concentration here reported [29,46,47].

The variation in the composition of essential oils due to the origin of the plants, the climate, and the extraction method is one of the characteristics of both essential oils and phytochemical extracts [31,42,48]. In the leaf samples collected for this work in different seasons, the variability in the EO composition could be attributed to metabolic changes [47,49]. However, in the case of extracts obtained from the same batch of leaves, but subjected to different treatments (i.e., milling and dehydration), differences in the composition of the extracts could be due to the treatment applied, as reported by Franco-Vega et al. (2016) [50]. In the present work, the compounds (−)-isoaromadendrene-(V), cis-caryophyllene, (−)-aristolene, β-germacrene, and (−)-spathulenol were only identified in the EO extracted from the *C. odorata* fresh leaves, while the compounds alloaromadendrene, β-cubebene, β-gurjunene, γ-elemene, α-selinene, (-)β-elemene, valencene, and cubenol were found in the EO extracted from the dehydrated leaves (Appendix A).

One additional characteristic of *C. odorata* EO that has attracted attention is the number of isomers of a single compound, which can be corroborated in the list of metabolites presented in Table 2 [30]. The essential oil of the neem tree was used as a positive control in the present work: its chromatographic analysis revealed a high content of terpenes, as shown in Appendix A.

### 2.3. Nanoemulsification of the Essential oil of Cedrela odorata in Water

#### 2.3.1. Selection of the Surfactant HLB

The emulsification of the *C. odorata* EO with the binary mixtures of ethoxylated sorbitan monooleate (Tween^®^ 80, TW80) and ethoxylated sorbitan monooleate (Span^®^ 20, SP20), at a Hydrophilic–Lipophilic Balance (HLB) between 8.98 and 12.89, generated the emulsion textures presented in Figure 2 and the cremation indices (CI) shown in Table 3 [51]. The emulsion textures were viscous and their color evolved from milky white to beige as the concentration of the TW80 surfactant increased from 0.28 to 2.03 in relation to the concentration of the SP20 surfactant. Meher et al. (2013) [52], who studied the stability of an emulsion based on its CI, reported that low CI values allow the formation of stable emulsions. Based on these reports and the results obtained in the present study, the HLB of 10.46, for a (surfactant mixture)/(essential oil) ratio (SM:EO ratio) equal to 1.5 (SOR = 1.5), was selected as the most adequate to prepare the desired nanoemulsion of the *C. odorata* EO. The HLB of 10.78 could also have been selected; however, the 10.46 value was preferred, considering that a small variation in the TW80 concentration will affect the CI to a lesser extent, if the HLB varies between 10.01 and 10.78, than if it varies between 10.46 and 11.03 [53]. Similar results to those obtained here were reported by Bachynsky et al. (1997) [54] and Matsaridou et al. (2012) [55]. These authors found that an HLB between 10 and 15 is suitable for preparing stable oil-in-water (O/W) nanoemulsions.

#### 2.3.2. Non-Ionic Surfactants Selected to Emulsify the Cedrela odorata Essential Oil

Emulsification of the *C. odorata* EO was carried out with binary mixtures of HLB = 10.46 prepared with non-ionic surfactants (TW80, TW20, SP80, SP20, and BL4), and the following components: EO, surfactants, and water at a ratio of 10:15:75. The evaluation of the different mixtures of surfactant agents was carried out in order to select the most suitable mixture to encapsulate the *C. odorata* oil [53]. Surfactants TW80, TW20, SP80, and SP20 are recommended for making emulsions using high-energy methods, and BL4 is suggested when low-energy methods are applied [22]. The HLB of these surfactants corresponds to those suggested for preparing emulsions of the O/W type, or W/O, as in the case of the SP20. Figure 3 shows the appearance of the emulsions formed during the two stages of the process implemented to develop the nanoemulsion. The letter A identifies the emulsions formed in one stage (magnetic stirring with heating), while the letter B represents the emulsions formed in two stages (magnetic stirring with heating and ultrasound in cycles of 2 min). The surfactant mixtures generated emulsions with the following characteristics.

TW80–SP80: in the first stage, an emulsion with a fluid appearance and a whitish color was generated. The application of ultrasound in three cycles of 2 min each generated a transparent emulsion with a bluish hue; the fourth and fifth cycles of ultrasound did not change the appearance of the emulsion.

TW80–SP20: during the first stage, an emulsion with a milky-beige appearance was generated. Its subsequent dispersion with ultrasound in 5 cycles of 2 min only changed the color of the emulsion from beige to milky white. According to McClements (2012) [56], milky emulsions have a droplet size of micrometric magnitude. The difference between the SP20 surfactant and the SP80 (the surfactant that allowed the nanoemulsion to be formed) is the unsaturated carbon chain in the hydrophobic fraction of the SP80; that is, a spatial arrangement of carbons that favors the alignment of SP80 molecules in the oil–water interface [57].

TW20–SP20: this mixture of surfactants produced a whitish dense and viscous emulsion. The high viscosity favored the adherence of the fluid to the vial walls. Ultrasonic mixing only generated foam. The lower performance of the TW20–SP20 mixture to emulsify the essential oil of *C. odorata* can be also attributed to the short chain of saturated carbons in the hydrophobic fractions of both surfactants.

TW80–BL4: this surfactant mixture generated an unstable emulsion with a milky and creamy appearance. The application of ultrasound led to the separation of phases. The phase separation is attributed to an overdispersion that could have favored the production of misshapen micelles within a wide range of diameters [58].

The emulsion produced by the mixture of the TW80 and SP80 surfactants developed the appearance of the desired nanoemulsion; that is, a transparent fluid with a bluish hue [58,59]. TW80 is an unsaturated fatty acid with double bonds in nonpolar chains, whose function is to promote the formation of O/W-type emulsions [58]. The HLB of SP80 is of the hydrophilic type, and its function is to stabilize the nanoparticles in the presence of ethoxylated sorbitan esters [60]. Mixtures of TW80 and SP80 surfactants have been successfully used to prepare nanoemulsions with EOs extracted from clove [58], canola [61], and leaves of *Illicium verum* Hook. f. plants Schisandraceae and *Rosmarinus officinalis* [62]. In such nanoemulsions, the surfactant mixture was added in different percentages, but the HLB values were in the range 9–16.7. These HLB values suggest the possibility of using other surfactants in the emulsification of the essential oil of *C. odorata*. However, regarding the mixtures of surfactants evaluated in the present study, it is a fact that their composition, and even the surfactant binary combinations, must be changed. Liu et al. (2021) [63], for example, reported the development of a stable PCM (phase change material) nanoemulsion by using a mixture of BL4–Tween 60 surfactants in a 6:4 ratio. The particle diameter was 60 nm.

Given that the TW80–SP80 mixture of HLB 10.46 allowed the formation of an emulsion with an appearance close to the nanoemulsion, this was selected to adjust the EO:SM:water ratio to be used in the production of the target nanoemulsion to be tested against *S. frugiperda*.

#### 2.3.3. Adjustment of the Ratio among the Cedrela odorata Essential Oil, a Mixture of Surfactants, and Water

The *C. odorata* EO was added in concentrations of 2.5% and 5%, and the TW80–SP80 surfactant mixture in the 1:1, 2:1, and 3:1 ratios (SM:EO = SOR) with respect to the maximum concentration of the EO used. The emulsions prepared with the surfactant mixture and the EO in a 2:1 ratio developed the characteristic appearance of a nanoemulsion, as can be seen in Figure 4. The droplet diameters of the emulsions 2.5:5:92.5 and 5:10:85 were 37.58 nm (PDI = 0.213) and 56.05 nm (PDI = 0.596), respectively (Figure 5). The effect of the SOR and the mixing conditions on the droplet diameter of a nanoemulsion prepared with lemon oil and TW80 as an emulsifying agent was studied by Rao and McClements (2011) [64]. These authors found that, within the 1 < SOR < 2 range and with the integration of the emulsion components via stirring and heating at 20–90 °C, the formation of a stable nanoemulsion (droplet diameter < 100 nm) is possible. The SOR in the 2–3 range was evaluated by Mazarei and Rafati (2019) [57] to emulsify the carvacrol-rich *Satureja khuzestanica* EO using different mixtures of surfactant agents. The TW80 and SP80 surfactant mixture of HLB = 10, added in a 3:1 ratio in relation to the EO studied, allowed them to produce a stable nanoemulsion with a droplet diameter equal to 95 nm. The results obtained in the present study are consistent with those previously reported by the aforementioned authors.

The smaller micelle diameter and lower PDI observed in the emulsion prepared with the different components at the 2.5:5:92.5 ratio were the criteria for its selection and application in the bioassays with *S. frugiperda*. The density and refractive index data used to calculate the droplet diameter of the nanoemulsion were 1.03 ± 0.003 g L^−1^ and 1.3458 ± 0.0001, respectively.

### 2.4. Nanoemulsion Stability of the Cedrela odorata Essential Oil at 2.5%

The ANOVA of the effect of time and storage temperature showed a significant effect on the drop diameter (F_time_ = 75.18, df = 5, *p* < 0.001; F_temperature_ = 112.78, df = 1, *p* < 0.001; F_(time,temperature)_ = 23.14, df = 5, *p* < 0.001) and the polydispersity index (PDL) (F_time_ = 29.92, df = 5, *p* < 0.001; F_temperature_ = 11.41, df = 1, *p* < 0.001; F_(time,temperature)_ = 4.35, df = 5, *p* < 0.001) of the 2.5% nanoemulsion of the *C. odorata* EO (Table 4, Appendix A). At a temperature of 5 ± 2 °C, the droplet diameter of the nanoemulsion increased by 60% during 30 days of storage but remained at the same order of magnitude at the end of day 90. In the case of the nanoemulsion stored at 25 ± 2 °C, the droplet diameter increased by 110% on day 30, and by 172% on day 90; that is, a droplet 2.7 times larger than the diameter of the initially formed droplet was obtained. The increase in droplet diameter is explained by the Ostwald maturation phenomenon and the low droplet growth rate under refrigeration conditions is attributed to the Lifshitz–Slezov–Wagner theory, regarding the inverse correlation between droplet growth due to Ostwald ripening and absolute temperature [58,65]. Although the droplet diameter increased, the diameter of less than 200 nm at the end of 90 days of storage is an indicator of the stability of the prepared emulsion, which in addition, was corroborated by the translucent blue appearance of the samples [66]. Of the two storage temperatures evaluated, refrigeration was assumed to be the best option for the conservation of the nanoemulsified formulation of the *C. odorata* EO.

The PDI of the *C. odorata* nanoemulsion was 0.227 ± 0.013 on day zero. According to Bernardi et al. (2011) [66] and Díaz-Blancas et al. (2016) [67], a PDI < 0.3 is representative of homogeneous and stable nanoemulsions. This homogeneity was favored by the HLB = 10.46 of the surfactant, as well as by the SOR = 2, as reported by Mazarei and Rafati (2019) [57]. The increase of 54% (30 days) and 58.5% (90 days) in the PDI of the *C. odorata* nanoemulsion kept at 5 ± 2 °C is an indicator of its stability at the end of 90 days of storage. In the case of the nanoemulsion stored at 25 ± 2 °C, the PDI increased by 81.7% and 94.4% in the time intervals of 30 and 90 days, respectively. This shows that the HLB of an SM is an important factor that controls the occurrence of instability phenomena, such as Ostwald ripening.

The micellar morphology of the *C. odorata* EO nanoemulsion stored at 5 ± 2 °C for 30 days is shown in Figure 6. The spherical shape is due to the low solubility of the *C. odorata* EO in water. It is important to highlight the larger droplet diameter obtained by transmission electron microscopy (TEM) compared with that measured by dynamic light scattering (DLS). However, such a difference has been attributed to the different lighting applied in both techniques. According to Silva da et al. (2018) [68] and Dal et al. (2016) [69], DLS analysis is more accurate for analyzing nanoparticles.

The turbidity and pH parameters of the 2.5% *C. odorata* EO nanoemulsion changed during storage at 25 °C and 5 °C. The largest changes, 71% (for turbidity) and 16% (for pH), were recorded in the nanoemulsions stored at room temperature, as can be seen in Table 4 and Appendix A. Turbidity is not an accurate indicator of stability. However, it is an important parameter, considering that fluid transparency is the qualitative criterion to identify a nanoemulsion [70]. Badawy et al. (2018) [71], who studied the stability of a nanoemulsion by the effect of pH, reported that the increase in the surface charge of the micelles promotes electrostatic repulsion between them, reducing the flocculation and dissolution of the nanoemulsion. In the present study, the pH value recorded in the nanoemulsions stored at 5 °C and 25 °C is consistent with the variation in diameter recorded during storage (Table 4).

The viscosity variation was negligible in the nanoemulsions stored at 5 °C. However, concerning those kept at 25 ± 2 °C, an increase from 0.896 ± 0.125 to 1.23 ± 0.27 cP was detected on day 90. An increase in viscosity in the O/W-type nanoemulsion was reported by Khan et al. (2011) [72] to be an indicator of floc formation. However, in the present study, no flocs were observed in the nanoemulsion. A verification of the elaboration of an O/W nanoemulsion is presented in Appendix A. The low viscosity values measured also confirm that an oil-in-water type nanoemulsion was produced [73].

### 2.5. Larvicidal Activity of the Cedrela odorata Essential Oil

The mortality percentage of the *C. odorata* EO against *S. frugiperda* first-instar larvae is presented in Figure 7, and the 50% lethal concentration (LC_50_) is shown in Table 5. The lethality of the oil increased significantly (F = 199.8; df = 5; *p* < 0.0001) from 27 ± 6% to 100% when increasing its concentration in the range 5000–40000 mg L^−1^. Compared to the lethality generated by the neem EO (positive control), no significant differences were observed (F = 0.19; df = 1; *p* = 0.662), although the higher lethality of the latter (>90%) from the concentration of 20,000 mg L^−1^ is worth noting. The nanoemulsification of the *C. odorata* EO with the TW80–SP80 mixture of surfactants did not generate the expected larvicidal activity, given that its LC_50_ decreased by 29% in comparison with that obtained with the EO in solution. However, despite the loss in larvicidal activity observed in the *C. odorata* nanoemulsion, it is important to consider the advantages of the nanoemulsified formulations with regard to their greater permanence in nature and long period stability, and the fact that they are environmentally friendly [58].

A decrease in the larvicidal activity of EOs nanoemulsified using ultrasound methods was reported by Cimino et al. (2021) [15] and was attributed to possible changes in its composition. Concerning the nanoemulsion of the *C. odorata* EO tested here, chromatographic analysis showed a decrease of 31% and 22% in their main compounds germacrene D and caryophyllene, respectively (Figure 8). This decrease can be attributed to their evaporation when the emulsion was treated with ultrasound, even though this operation was carried out in an ice bath and in cycles of 2 min. Germacrene D and caryophyllene are two of the metabolites extracted from plants that have shown insecticidal activity against *S. frugiperda* [74,75]. Therefore, a decrease in their composition could have affected the insecticidal activity of the nanoemulsion. The ratio of the heights of the corresponding peaks in the two chromatograms shown in Figure 8 allows us to clearly see the change in the composition of the essential oil of *C. odorata* before and after the ultrasound treatment. Despite the change in the composition of the oil in the nanoemulsion, it is important to highlight its remaining larvicidal activity, which produced the mortality percentages shown in Figure 7. In the chromatogram shown in Figure 8A, the peak in temperature 40.16 could be traces of Tween 80 (Appendix A).

The exposure of *S. frugiperda* larvae to the neem EO nanoemulsion (positive control) at a concentration of 25,000 mg L^−1^ (the maximum concentration of the *C. odorata* EO nanoemulsion) generated a mortality rate as high as that obtained from its corresponding EO applied in solution with acetone (Figure 7). These results could be explained by the high boiling temperature of the identified fatty acids, greater than 230 °C, and even greater than 750 °C in the case of azadirachtin, a compound associated with the neem tree. Zorga et al. (2020) [76] compared different extraction methods of essential oil from oats and found that ultrasonic extraction allows a preferential extraction of limonenes, and azadirachtin is included in this group.

Finally, it must be stated that, of the different sesquiterpene compounds identified in the *C. odorata* EO, germacrene D, caryophyllene, caryophyllene oxide, α-caryophyllene, and γ-elemene have toxic properties against insects of the Lepidoptera genus. Kiran et al. (2006) [77] determined an LC_50_ of 24.83 mg kg^−1^ against *Spodoptera litura* exposed to germacrene D isolated from *Chloroxylon swietenia* DC. Cárdenas-Ortega et al. (2015) [78] reported mortality rates of 75% and 65% in *S. frugiperda* larvae exposed to caryophyllene and caryophyllene oxide, respectively, at a concentration of 80 µg mL^−1^. The sesquiterpenes α-caryophyllene and γ-elemene were extracted from *Wedelia prostrata* by Benelli et al. (2018) [79]. The LD_50_ against *Spodoptera litura* reported by these authors was 12.89 μg mL^−1^ and 10.64 μg mL^−1^, respectively. The lethality of these compounds is high; however, it is important to mention that in nanoemulsion, the release of the active agent must be taken into consideration for comparison purposes.

Neem EO is one of the most widely used botanical pesticides for insect pest control [37]. The similar performance of the *C. odorata* EO compared with neem oil indicated that the *C. odorata* EO could have good potential for commercial use in the biological control of insect pests. Information on the insecticidal properties of limonoids extracted from plants of the *Cedrela* genus was reported by Céspedes et al. (2000) [35]. However, the development of a nanoemulsified formulation of the *C. odorata* EO has not been reported. The present work was oriented to the development of a nanoemulsion of the essential oil extracted from dehydrated leaves of *C. odorata*. The insecticidal activity of the nanoemulsion against *S. frugiperda* was lower than that obtained with the EO in a ketone solution. However, it is important to consider the advantages of using nanoemulsions; that is, their greater permanence in the environment, their greater effectiveness in ensuring a homogeneous dispersion of the nanoemulsion in the plant, and that they are environmentally friendly, considering that water is used as a dispersant instead of solvents. Furthermore, it is important to mention the greater larvicidal activity of the EO when compared with the ethanolic extract obtained from the same plant [36].

## 3. Materials and Methods

### 3.1. Plant Material

Leaves of *C. odorata* (red cedar or American cedar) were collected from randomly selected trees in Santos Reyes Nopala, Oaxaca, Mexico (located at 16°04′24.0′′ N and 97°07′04.1′′ W) in June (batch 1) and September (batch 2) of 2018. The plant was identified by M.Sc. Ernestina Cedillo Portugal at the Autonomous University of Chapingo, Mexico. The plant samples were deposited at the Jorge Espinosa Salas herbarium at the same university with the registration number 25,946 (*Cedrela odorata* L.).

One-third of the leaves from the first batch was deposited into plastic bags for storage at −76 °C (Thermo Scientific Ultra Freezer, Model 9131 TSE320A, Waltham, MA, USA); this batch was identified as fresh *C. odorata* (FL) leaves. The remaining leaves were dehydrated in the shade at room temperature; this batch was named *C. odorata* dehydrated leaves (HD). The fresh and dehydrated leaves were ground in a blender (Oster Model BPST02-B00-013 Newell Brands, Atlanta, Georgia, USA) to a particle size of 35 mesh; the fresh leaves were frozen with liquid nitrogen before grinding. The powder was stored in black plastic bags at −76 °C (Thermo Scientific Ultra-Freezer, Model 9131 TSE320A, Waltham, MA, USA) until its later use in the extraction of its EO. All leaves from the second batch were dried, ground, and stored deep-frozen (−76 °C) as described above.

### 3.2. Extraction of the Cedrela odorata Essential Oil

The EO was extracted by hydrodistillation in a Clevenger apparatus of 3 L [80]. The extraction was conducted in batches of 100 g of plant powder soaked in 1 L of deionized water; the process lasted 3 h at water boiling temperature [81]. The EO accumulated in the Clevenger trap was placed in a 10 mL beaker, where the free water humidity was removed with 1 g of anhydrous sodium sulfate (J.T.Baker: 3891-01). The solid–liquid mixture was filtered through a cotton ball covered with anhydrous sodium sulfate; the EO adhering to the filter was recovered by dragging with HPLC-grade dichloromethane (Honeywell: 34856). Dichloromethane was removed by rotary evaporation and the EO was collected in a 4 mL amber vial [82]. The vial was stored at 4 °C until the next use.

The EO yield (*Rd*) was determined using the following equation:(1)$$Rd=\left(\;\frac{p_{1}}{p_{2}}\;\right)100$$
where *p*_1_ is the EO weight extracted from the plant powder and *p*_2_ represents the weight of plant powder used in the extraction process (100 g) [83].

The EO was characterized by its density (ρ = m/V; m (g) and V (mL), mass and volume of the EO), refractive index (Abbe refractometer model 10450), and chemical composition. The chemical composition was determined by gas chromatography coupled with mass spectrometry (GC–MS). This was performed (liquid samples of 3 μL) on a PerkinElmer Clarus 580/Clarus SQ8S implemented with a 30 m × 0.32 mm × 0.25 μm Elite-5MS capillary column (PerkinElmer, Waltham, MA, USA) (1,4-bis(dimethylsiloxy)phenylene dimethylpolysiloxane) [84]. The GC–MS was heated from 40 °C to 260 °C at a rate of 5 °C min^−1^; the temperature in the injector was 250 °C. Helium was used as the carrier gas at 0.8 mL min^−1^ and the ionization energy was 70 eV. The mass range for MS was 15–500 m/z with a transition temperature of 230 °C and a source temperature of 200 °C. The NIST library version 6.0.0 was used as a reference for the tentative identification of the major compounds of the *C. odorata* EO. Results reported by other authors were also checked for comparison purposes.

### 3.3. Nanoemulsified Formulation of the C. odorata Essential Oil

The nanoemulsion was developed in three stages. In the first, the most suitable *HLB* to emulsify the *C. odorata* EO was determined. This activity was conducted with different mixtures of the ethoxylated sorbitan monooleate (Tween^®^ 80 (TW80), Sigma Aldrich (San Luis, MO, USA): P1754, HLB = 15) and sorbitan monolaurate (Span^®^ 20 (SP20), Sigma Aldrich (San Luis, MO, USA): S6635, *HLB* = 8.6) surfactants. The *HLB* was set in the range 8.99–12.89 (Table 6) and was determined from the following equation:(2)$$HLB=\frac{m_{TW80}\;\times \;HLB_{TW80\;}+m_{SP20}\;\times \;HLB_{SP20}}{m_{TW80}+m_{SP20}}$$
where *m_TW_*_80_ and *m_SP_*_20_ are the mass of the TW80 and SP20 surfactants, respectively, while the variables *HLB_TW_*_80_ and *HLB_SP_*_20_ are their corresponding *HLB* values [58]. The *C. odorata* EO, the surfactant mixture, and the distilled water were added in a 10:15:75 ratio, according to that reported by Gómez-Vega (2014) [85] and Alcántara Martínez et al. (2016) [86].

In the second stage, the surfactants (two in a binary mixture) that allowed the *C. odorata* EO to be dispersed with a droplet diameter of less than 200 nm were selected. The HLB of the surfactant mixture was that determined in stage 1 and with the EO/surfactant ratio of 1.5. The non-ionic surfactants used were as follows: sorbitan monooleate (Span^®^ 80 (SP80), Fluka: 85548, *HLB* = 4.3); ethoxylated sorbitan monooleate (Tween^®^ 80 (TW80), Sigma Aldrich: P1754, *HLB* = 15); sorbitan monolaurate (Span^®^ 20 (SP20), Sigma Aldrich: S6635, *HLB* = 8.6); polyoxyethylene sorbitan monolaurate (Tween^®^ 20 (TW20), Sigma Aldrich: P1379, *HLB* = 16.7); and polyethylene glycol dodecyl ether (Brij^®^ L4 (BL4), Sigma Aldrich: 235989-1L, *HLB* = 9).

The third stage included a verification of the most suitable surfactant/EO ratio (SOR) to form a stable nanoemulsion. The tested SOR values were: 1, 1.5, 2, and 3. The HLB of the surfactant mixture was that determined in stage 1. Nanoemulsions were prepared with the surfactants selected in stage 2.

The emulsions (or nanoemulsions) of the different stages were prepared in 15 g batches in 100 mL bottles [85,86]. Surfactants previously weighed on an analytical balance (OHAUS Explorer Pro EP214C, Parsippany, NJ, USA) were mixed for 10 min with stirring at 1000 rpm, at a controlled temperature of 40–45 °C (Mr Hei-Tec, Heidolph, Schwabach, Germany). Subsequently, the *C. odorata* EO was added slowly for its incorporation into the mixture of surfactants for 10 min; this operation was conducted under the same stirring and temperature conditions. The water was integrated into the oily mixture at 1400 rpm at a controlled temperature of 73–75 °C; this operation lasted 30 min. The emulsion formed was analyzed with regard to the phase number, turbidity, and color. Milky emulsions, or emulsions with separate phases, were ultrasonicated (Hielscher model UP200Ht ultrasound implemented with a 7 mm sonotrode, Teltow, Germany) in cycles of 2 min until a transparent bluish emulsion was observed [87]. More than five cycles was an indicator that the nanoemulsion would not form.

The emulsions (or nanoemulsions) were centrifuged at 1500 rpm for 30 min [52], and the emulsion stability, as a cremation index (%*CI*), was determined from the following equation:(3)$$\%CI=\frac{CLH}{ETH}\times 100$$
where *CLH* is the height of the cream layer and *ETH* denotes the total height of the fluid.

The sensitivity of the surfactant *HLB* of the formed nanoemulsion was determined by changing the composition of the mixture of surfactants by ±0.3 g [88]. The nanoemulsions were prepared in triplicate and the physicochemical properties were determined for each one.

### 3.4. Nanoemulsion Characterization

The formation of an oil-in-water (*O*/*W*) nanoemulsion was verified by a staining and solubility test [72,89]. The staining test consisted of placing a drop of dye in solution (HYCEL crystal violet: 42555) next to a drop of the prepared nanoemulsion. The fusion of the two drops indicates that the nanoemulsion is of the *O*/*W* type. For the solubility test, 1 mL of the nanoemulsion was mixed for 30 min with 1 mL of distilled water. The dilution of the nanoemulsion in a single-phase fluid indicates that the nanoemulsion is of the *O*/*W* type. If two phases are formed, it is a water-in-oil (*W*/*O*)-type nanoemulsion.

The stability of the nanoemulsion in samples stored for 90 days at room temperature (25 ± 2 °C) and in refrigeration (5 ± 2 °C) was determined according to Isaac et al. (2008) [88] and Badawy et al. (2018) [71]. This test comprised a determination of the nanoemulsion appearance (phase number, color, and turbidity) and droplet morphology, as well as a measurement of the droplet size and polydispersity. The dropping size and polydispersity were determined by dynamic light scattering (DLS) using 1 mL samples in a Malvern Zetasizer Nano ZS90 (Malvern Instruments Ltd., Worcestershire, UK). The measurements were taken at room temperature, and each reading was done in triplicate [71].

The morphological analysis of the nanoemulsions was performed using transmission electron microscopy (TEM) [90]. Ten microliters of samples were adsorbed for 2 min on copper grids, then negatively stained with filtered phosphotungstic acid adjusted to pH 7.0 for 1 min [91]. The excess liquid of the samples was dried with Whatman filter paper and the samples were directly observed in the TEM JEM-1400 (JEOL, Peabody, MA, USA).

The pH measurement was performed with 10 mL samples using a digital pH meter (pH spear, Eutech, Oakton, Vernon Hills, IL, USA). The percentage of turbidity was analyzed using a UV-visible spectrophotometer (Thermo Scientific GENESYS 10UV UV-Vis, Madison, WI, USA) at a wavelength of 600 nm [92]. Distilled water was used as a blank. The viscosity (centipoises, cp) was recorded in a digital vibro MODEL SV-10 viscometer (Tokio, Japan) in 30 mL samples.

The characterization of the nanoemulsion was performed in triplicate.

### 3.5. Larvicidal Activity of the Cedrela odorata Essential Oil

The larvicidal activity of the EO was conducted in vitro using bioassays with first-instar larvae of *S. frugiperda* obtained from the Biotechnology Research Center of the Autonomous University of the State of Morelos, Mexico. The bioassays were set up in polystyrene plates (Mexico Thermo Scientific Nunc; 24-well plates of 1 mL and an opened surface of 1.9/cm^2^), half-filled with a sterile synthetic diet [36]. The EO was dissolved in acetone and poured over the gelled diet in aliquots of 35 μL at concentrations in the range of 10,000–250,000 mg L^−1^. After the evaporation of the solvent, one *S. frugiperda* first-instar larva was placed into each well [93]. The polystyrene plates were covered with plastic film and perforated to allow fresh air to enter the wells. Larvae were reared at 25 ± 2 °C, in relative humidity of 80 ± 5%, and for an 18/6 photoperiod. Dead and surviving larvae were recorded every 24 h for 7 days. The weight (OHAUS Explorer brand analytical balance) and size (Vernier; Scale Regla Dial Caliper mm metric) of the surviving larvae were recorded on the seventh day. Distilled water was used as a negative control and neem oil (IBCER S.A. of C.V. Sinaloa, Mexico) as a positive control in concentrations of 5000, 20,000, 50,000, and 100,000 mg L^−1^. The treatments were implemented in quadruple with 12 larvae per replicate, and 48 larvae per concentration and per treatment.

### 3.6. Larvicidal Activity of the Cedrela odorata Essential Oil Nanoemulsified Formulation

Larvicidal activity was determined as described in Section 2.5. The nanoemulsion was added at concentrations of 1250, 2500, 6250, and 12,500 ppm, which were prepared by dilution with distilled water from a stock solution of 2.5% concentration (25,000 ppm). A neem oil nanoemulsion (IBCER S.A. de C.V., Sinaloa, México) was used as a positive control [94]. Two negative controls were implemented, one with distilled water and the other with the surfactant mixture used in the preparation of the nanoemulsion.

### 3.7. Statistical Analysis

A mixed experimental design was implemented to study the larvicidal effect of the *C. odorata* EO (solubilized in acetone and emulsified with a mixture of surfactants), to which *S. frugiperda* was exposed at different concentrations. The reported values are the mean of the results obtained from four replicates. The difference between means was determined from a mixed ANOVA and the Bonferroni test for *p* < 0.05. The 50% lethal concentration (LC_50_) of the essential oil of *C. odorata* was determined using the probit method. PASW Statistics version 18.0.0 USA (2009) was used for statistical analysis.

## 4. Conclusions

*C. odorata* is a plant cultivated for the production of wood, from which large volumes of waste, mainly leaves and branches, are produced. The GC-MS analysis of the EO extracted from *C. odorata* dehydrated leaves revealed that it comprises mainly sesquiterpenes, with caryophyllene and germacrene D predominating. The *C. odorata* EO in solution and in nanoemulsified formulation presented larvicidal activity against *S. frugiperda*. The EO larvicidal activity detected was similar to that obtained with neem EO, which is already commercialized for application in pest insect control. The insolubility in water, volatility, and easy degradation of EO are properties that have been overcome by their encapsulation in nanoemulsions. In the present work, the nanoemulsification of the *C. odorata* EO was successfully obtained with a mixture of TW80 and SP80 surfactants. The emulsion stability of the *C. odorata* EO was verified in samples stored at 5 °C and 25 °C for 90 days.

The insecticidal properties of the EO extracted from *C. odorata* waste should be considered for use in the production of sustainable insecticides. Not only would the problem of the depletion of agricultural crops caused by *S. frugiperda* be alleviated, but also waste that would otherwise contribute to environmental pollution could be used.

## Figures and Tables

**Figure 1 molecules-27-02975-f001:**
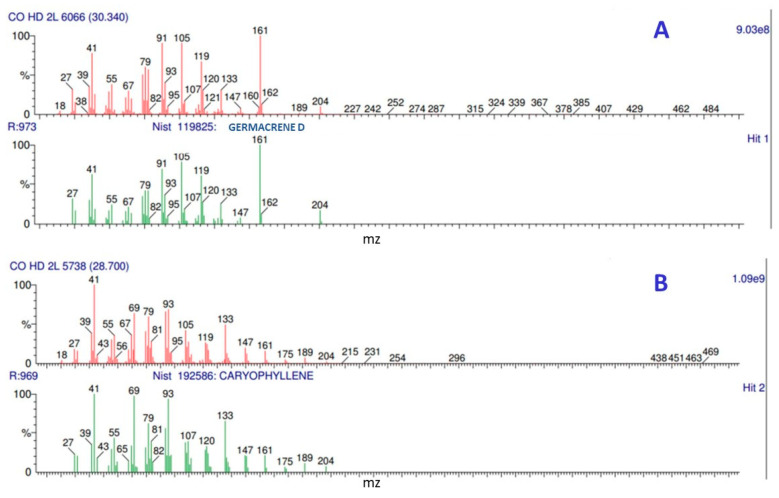
Mass spectrum of the major compounds identified in the *Cedrela odorata* essential oil. identification with the mass spectrum of the NIST library. (**A**) Germacrene D, identification with an approximation of 97.3%. (**B**) Caryophyllene, identification with an approximation of 96.7%.

**Figure 2 molecules-27-02975-f002:**
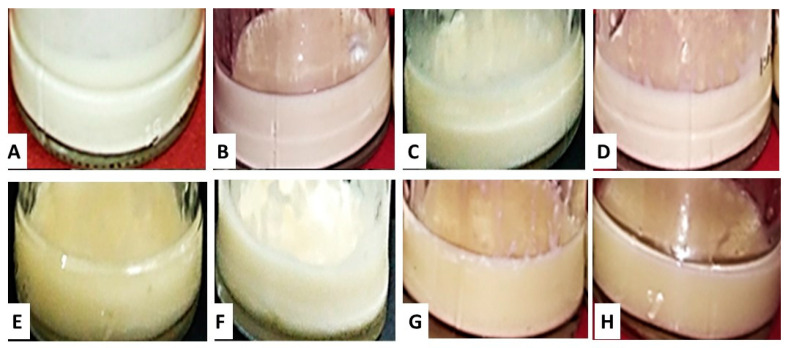
Texture of the *Cedrela odorata* essential oil (10%) and water (75%) mixture after their emulsification with different mixtures (15%) of the surfactants Tween 80 and Span 20. Hydrophilic–Lipophilic Balance (HLB) value of the surfactant mixtures: (**A**) 8.98; (**B**) 10.01; (**C**) 10.46; (**D**) 10.78; (**E**) 11.03; (**F**) 11.48; (**G**) 12.06; and (**H**) 12.89.

**Figure 3 molecules-27-02975-f003:**
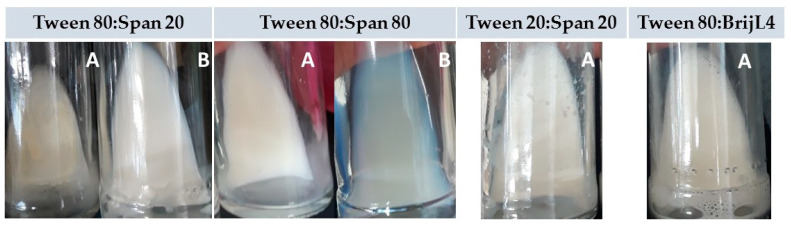
Emulsions of the *Cedrela odorata* essential oil with different binary mixtures of surfactants of HLB = 10.46 and a 10:15:75 (essential oil:surfactant:water) ratio: (A) emulsions prepared with magnetic stirring and heating; and (B) dispersion of oil droplets using ultrasound in cycles of 2 min.

**Figure 4 molecules-27-02975-f004:**
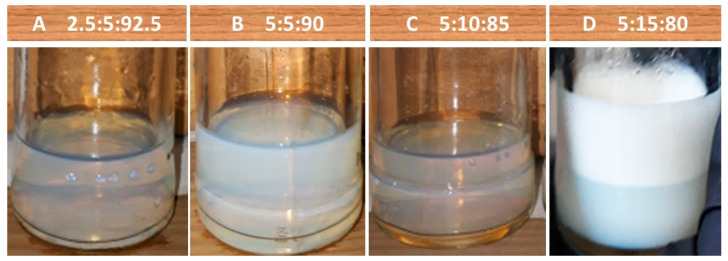
Emulsions of the *Cedrela odorata* essential oil in terms of the essential oil:surfactant mixture:water ratio: (**A**) 2.5:5:92.5; (**B**) 5:5:90; (**C**) 5:10:85; and (**D**) 5:15:80. Surfactant mixture (Tween 80–Span 80) of HLB = 10.46.

**Figure 5 molecules-27-02975-f005:**
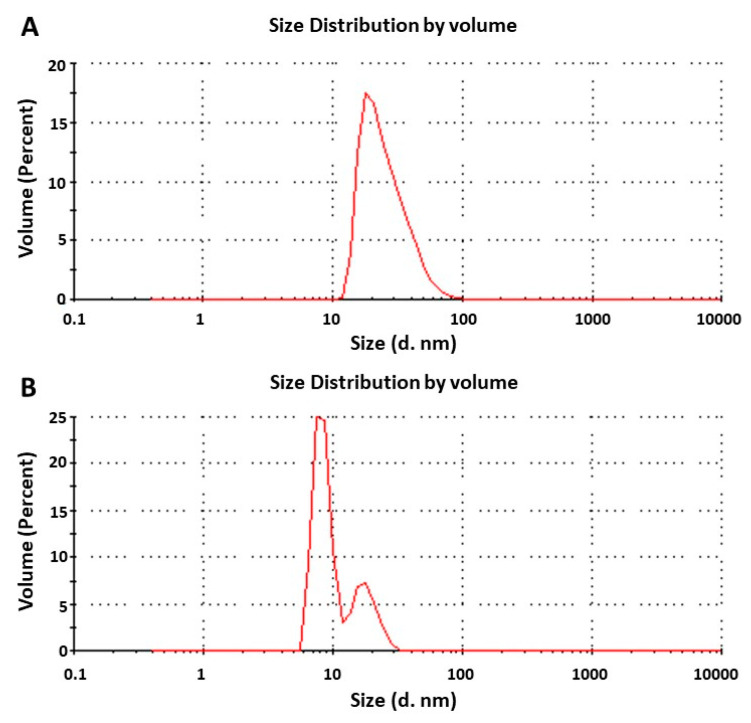
Dynamic light scattering (DLS) diagrams of the droplet diameter in the nanoemulsions: (**A**) 2.5:5:92.5; and (**B**) 5:10:85.

**Figure 6 molecules-27-02975-f006:**
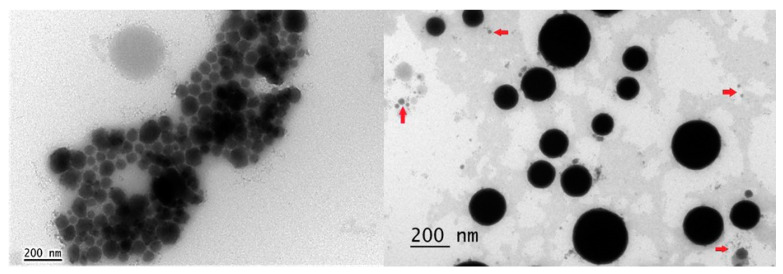
Transmission electron microscopy (TEM) morphology of the micelles in the nanoemulsion of the *Cedrela odorata* essential oil. The red arrows show the predominant droplet diameter in the DLS analysis at day 30 in samples stored at 5 ± 2 °C.

**Figure 7 molecules-27-02975-f007:**
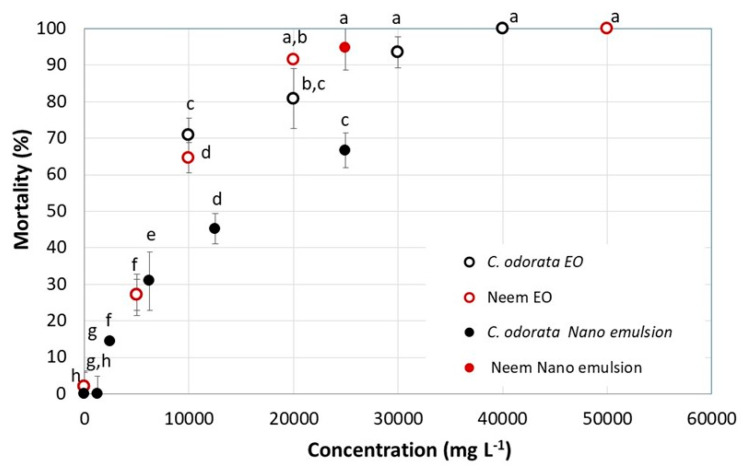
Mortality (mean (%) ± SE) of first-instar larvae of *Spodoptera frugiperda* measured by the effect of the concentration (mg kg^−1^) of the *Cedrela odorata* essential oil, alone or in a nanoemulsified formulation. Mortality was recorded on day seven. Positive control: neem essential oil; negative control: Tween 80–Span 80 surfactant mixture. Two-way ANOVA. Dots with the same letter (a–h) do not significantly differ (Bonferroni test for an α = 0.05).

**Figure 8 molecules-27-02975-f008:**
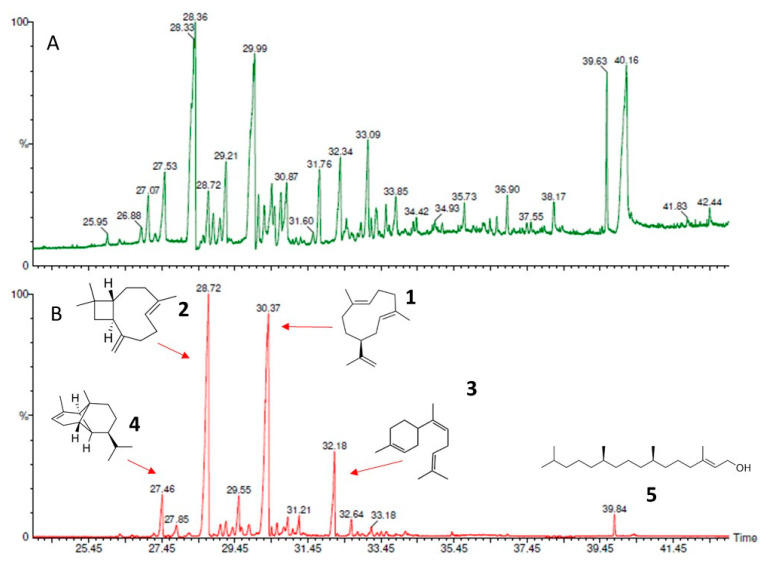
Chromatograms of the *Cedrela odorata* essential oil: (**A**) oil extracted with dichloromethane from the nanoemulsion of the *C. odorata* essential oil at 2.5%; and (**B**) oil extracted by hydrodistillation from the *C. odorata* dehydrated leaves. Major components: 1 = germacrene D; 2 = caryophyllene; 3 = cis-α-bisabolene; and 4 = copaene, 5 = phytol.

**Table 1 molecules-27-02975-t001:** Characteristics of the essential oil extracted from the *Cedrela odorata* dehydrated leaves. Data are the average of *n* = 3 and are expressed as mean ± standard deviation. The leaf batch was collected in June 2018.

Leaf Parameter	Dehydrated Leaf	Fresh Leaf
Color	Greenish-yellow	Greenish-yellow
Odor	Dry leaf	Dry leaf
Density (g mL^−1^)	0.91 ± 0.01	0.90 ± 0.018
Refractive index	1.5 ± 0.01	1.5 ± 0.01
Yield (%)	0.58 ± 0.13 ^1^	0.26 ± 0.04

^1^ Yield of the leaf batch collected in September 2018, 0.44 ± 0.04%.

**Table 2 molecules-27-02975-t002:** Compounds (≥1%) were extracted by hydrodistillation from the fresh (FL) and dehydrated (DL) leaves of *Cedrela odorata*.

Peak	Retention Time (min)	Compound	FL (%) June 2018	DL (%) June 2018	DL (%) September 2018
1	27.589	copaene	3.75	8.17	1.83
2	27.955	β-elemene	1.49	3.25	3.23
3	28.315	cis-caryophyllene	-	1.92	-
4	28.825	caryophyllene	34.12	23.93	51.79
5	29.050	alloaromadendrene	1.12	-	3.04
6	29.200	β-cubebene	1.29	-	-
7	29.305	(−)-aristolene	-	1.78	-
8	29.385	β-gurjunene	0.73	-	1.65
9	29.550	α-caryophyllene	3.32	4.59	4.68
10	29.835	γ-elemene	1.16	-	-
11	30.456	germacrene D	36.85	25.24	9.92
12	30.605	α-selinene	0.89	-	1.62
13	30.696	β-germacrene	-	1.73	-
14	30.895	(-)β-elemene	1.44	-	1.8
15	30.976	(−)-isoaromadendrene-(V)	-	1.95	-
16	31.050	valencene	-	-	1.87
17	31.206	δ-cadinene	1.34	3.62	2.77
18	31.256	cis-α-bisabolene	7.30	8.83	3.16
19	32.636	caryophyllene oxide	1.23	2.72	1.97
20	33.356	cubenol	-	-	3.01
21	33.687	(−)-spathulenol	-	1.00	-
22	39.889	phytol	1.20	3.95	1.44
		% Total	97.23	92.68	93.78

**Table 3 molecules-27-02975-t003:** Creaming index of the *Cedrela odorata* essential oil emulsion from the effect of the Hydrophilic–Lipophilic Balance HLB of the Tween 80–Span 20 surfactant mixture. Essential oil/surfactant mixture/water = 10:15:75.

Tween 80	Span 20	HLB	%CI
6	94	8.98	50
22	78	10.01	4
29	71	10.46	0
34	66	10.78	0
38	62	11.03	10
45	55	11.48	20
54	46	12.06	24
67	33	12.89	24

**Table 4 molecules-27-02975-t004:** Nanoemulsion stability of the *Cedrela odorata* essential oil at 2.5%. Samples were stored at 25 ± 2 °C and 5 ± 2 °C for 90 days. Two-way MANOVA. Mean values (± SE) that do not share the same letter (^a^, ^b^, ^c^, ^d^, ^e^, ^f^, ^g^) are significantly different according to Tukey’s test (α = 0.05).

Temperature	Time (Days)	0	1	7	15	30	90
25 °C	Drop size, nm	38.9 ± 3.1 ^a^	39.4 ± 4.0 ^a^	60.2 ± 8.3 ^cd^	65.9 ± 13.5 ^d^	81.8 ± 9.1 ^e^	105.7 ± 9.4 ^f^
	PDL	0.227 ± 0.013 ^a^	0.235 ± 0.046 ^a^	0.339 ± 0.101 ^abcd^	0.459 ± 0.067 ^def^	0.559 ± 0.034 ^f^	0.479 ± 0.032 ^ef^
	Turbidity, %	34.3 ± 0.4 ^a^	36.8 ± 2.3 ^ab^	39.18 ± 2.5 ^abc^	46.3 ± 1.3 ^def^	50.9 ± 3.7 ^f^	58.8 ± 1.8 ^g^
	pH	6.3 ± 0.0 ^c^	6.3 ± 0.1 ^c^	6.3 ± 0.1 ^c^	6.2 ± 0.1 ^bc^	6.1 ± 0.0 ^b^	5.3 ± 0.4 ^a^
5 °C	Drop size, nm	38.9 ± 3.1 ^a^	42.1 ± 7.6 ^a^	43.4 ± 4.3 ^ab^	48.1 ± 10.1 ^abc^	62.3 ± 3.1 ^d^	56.2 ± 13.0 ^bcd^
	PDL 5 °C	0.227 ± 0.013 ^a^	0.285 ± 0.092 ^ab^	0.335 ± 0.072 ^abc^	0.345 ± 0.102 ^abcd^	0.412 ± 0.118 ^cde^	0.389 ± 0.129 ^bcde^
	Turbidity, %	34.3 ± 0.4 ^a^	39.3 ± 0.8 ^bc^	42.3 ± 5.1 ^cd^	42.6 ± 2.4 ^cd^	44.4 ± 2.5^d e^	47.7 ± 7.1 ^ef^
	pH	6.3 ± 0.0 ^c^	6.3 ± 0.0 ^c^	6.4 ± 0.1 ^c^	6.4 ± 0.1 ^c^	6.3 ± 0.1 ^c^	6.2 ± 0.1 ^bc^

**Table 5 molecules-27-02975-t005:** LC_50_ and LC_95_ of the *Cedrela odorata* essential oil applied in a ketone solution and a nanoemulsion against first-instar larvae of *Spodoptera frugiperda*.

	LC_50_ (mg kg^−1^)	Confidence Limits at 95% (mg kg^−1^)		LC_95_ (mg kg^−1^)	Confidence Limits at 95% (mg kg^−1^)		χ^2^	df	Sig.
		lower limit	upper limit		lower limit	upper limit			
Essential oil									
*C. odorata*	7712.1	6122.9	9214.9	30,043.6	23,652.1	42,938.7	9.573	18	0.945
Neem	7667.8	6355.5	8976.9	23,371.3	18,208.7	35,250.4	0.773	18	1.000
Nanoemulsion									
*C. odorata*	9952.4	7237.9	14,968.1	-	-	-	2.701	18	1.000

**Table 6 molecules-27-02975-t006:** Concentrations of the Tween 80 (TW80) and Span 20 (SP20) surfactant mixtures and their corresponding HLB.

Surfactant Mixture	TW80 (%)	SP (%)	*HLB*
SM 1	6	94	8.98
SM 2	22	78	10.01
SM 3	34	66	10.78
SM 4	54	46	12.06
SM 5	67	33	12.89

## Data Availability

The experimental data are already included in the article in the form of graphs and tables. Additional information will be provided upon request.

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
