# Peer review of "Nanoemulsified Formulation of Cedrela odorata Essential Oil and Its Larvicidal Effect against Spodoptera frugiperda (J.E. Smith)"

_molecules, 2022, doi:10.3390/molecules27092975_

Round 1
Reviewer 1 Report
The paper “Nanoemulsified Formulation of the Cedrela odorata Essential 2 Oil and its Larvicidal Effect against Spodoptera frugiperda (J.E. 3 Smith)” is focused to produce sustainable and effective insecticides in the fight against S. frugiperda.
The paper is interesting; however, the authors should broaden the introduction section and the discussion of the results, both are poor.
Introduction:
It would be appropriate to add more information than the technology used as a method of preparation, why a nano approach? (Zoxamide accumulation and retention evaluation after nanosuspension technology application in tomato plant, Corrias et al. 2021)
Why nanoemulsions were chosen?
Nanoemulsion characterization and stability:
It would be appropriate to schematize the characterization and stability with a table, it would make it clearer and more schematic.
Figure1 and 2 to be improved
Reviewer 2 Report
This manuscript treats constituent analysis of essential oil (EO) preparations from Cedrela odorata leaves, and the EO emulsion formulations prepared with combinations of Tween 80 (or 20), Span 80 (or 20), and/or Brij L4. The larvicidal effects of a formulation of Cedrela odorata EO on Spodoptera frugiperda, in comparison with the effects of Cedrela odorata EO preparation, a neem emulsion formulation, a neem EO preparation were also shown. Although the idea of the EO formulations for the usages of the insects seems to be interesting, the results shown in Figure 8 suggested that the EO solution in acetone is good enough relative to the emulsion formulation. The authors should indicate the superiority of the usage of the emulsion formation more clearly. The stability of the EO constituents in the soil should be given, too. Additionally, the significant digits of the data in Table 4 should also be considered.
Round 2
Reviewer 2 Report
The manuscript was revised adequately based on the comments by this reviewer, and now this reviewer recommends the acceptance of the manuscript as a paper of the journal Molecules.
Author Response
The authors appreciate the reviewer's comments on the manuscript molecules-1687530. Thank you very much.